# Adverse Neonatal Outcome of Pregnancies Complicated by Preeclampsia

**DOI:** 10.3390/biomedicines10082048

**Published:** 2022-08-22

**Authors:** Piotr Tousty, Magda Fraszczyk-Tousty, Joanna Ksel-Hryciów, Beata Łoniewska, Joanna Tousty, Sylwia Dzidek, Kaja Michalczyk, Ewa Kwiatkowska, Aneta Cymbaluk-Płoska, Andrzej Torbé, Sebastian Kwiatkowski

**Affiliations:** 1Department of Gynecology and Obstetrics, Pomeranian Medical University, 70-111 Szczecin, Poland; 2Department of Neonatology and Neonatal Intensive Care, Pomeranian Medical University, 70-111 Szczecin, Poland; 3Department of Gynecological Surgery and Gynecological Oncology of Adults and Adolescents, Pomeranian Medical University, 70-111 Szczecin, Poland; 4Department of Nephrology, Transplantology and Internal Medicine, Pomeranian Medical University, 70-111 Szczecin, Poland

**Keywords:** neonatal outcome, preeclampsia, angiogenesis markers, neonatal complications, preterm birth

## Abstract

Despite many available treatments, infants born to preeclamptic mothers continue to pose a serious clinical problem. The present study focuses on the evaluation of infants born to preeclamptic mothers for the occurrence of early-onset complications and attempts to link the clinical status of such infants to the angiogenesis markers in maternal blood (sFlt-1, PlGF). The study included 77 newborns and their mothers diagnosed with preeclampsia. The infants were assessed for their perinatal outcomes, with an emphasis on adverse neonatal outcomes such us infections, RDS, PDA, NEC, IVH, ROP, or BPD during the hospitalization period. The cutoff point was established using the ROC curve for the occurrence of any adverse neonatal outcome and it was 204 for the sFlt-1/PlGF and 32 birth week with AOC 0.644 and 0.91, respectively. The newborns born to mothers with high ratios had longer hospitalization times and, generally, were more frequently diagnosed with any of the aforementioned adverse neonatal outcomes. Also, the neonates born prior to or at 32 wkGA with higher sFlt-1/PlGF ratios were statistically significantly more common to be diagnosed with any of the adverse neonatal outcomes compared to those with lower ratio born prior to or at 32 wkGA. The sFlt-1/PlGF ratio can be a useful tool in predicting short-term adverse neonatal outcomes. Infants born after a full 33 weeks gestation developed almost no severe neonatal complications. Appropriate screening and preventive healthcare for preeclampsia can contribute significantly to reducing the incidence of neonatal complications.

## 1. Introduction

Preeclampsia (PE) is one of the pregnancy complications which, despite rapid developments in perinatology, continues to present major therapeutic challenges. It is a multi-factorial disorder that occurs in approximately 2–8% of pregnant women worldwide. Two types of the condition are distinguished: early-onset preeclampsia (before 34 weeks’ gestation (wkGA)) and late-onset preeclampsia (beyond 34 wkGA) [1,2].

The pathogenesis of preeclampsia is considered to include abnormal uterine artery remodeling and poor placental tissues differentiation resulting in oxidative stress in the placenta and vascular endothelial damage. In a developing preeclampsia, a distortion is observed of the proportions between angiogenic factors, such as vascular endothelial growth factor (VEGF) and placental growth factor (PlGF) on the one hand and anti-angiogenic factors (i.e., soluble fms-like tyrosine kinase-1(sFlt-1) and endoglin (sEng)) on the other. Elevated sFlt-1 and reduced PlGF and VEGF concentrations are then observed. The literature is abundant in reports comparing the values of the sFlt1/PlGF ratio with maternal and/or neonatal complications [3,4,5,6,7,8,9].

In today’s world, preeclampsia is the cause of many preterm births and perinatal mortality in women and neonates. There are many papers discussing the effects of preeclampsia on infants’ birth status and its impact on the occurrence of various preterm birth complications that develop in the neonate during hospitalization in a neonatal care unit [10,11,12,13]. First of all, it should be remembered that compared with children born to healthy mothers, these infants more often demonstrate complications such as body weight below the 10th percentile, neutropenia, thrombocytopenia, and complications typical of preterm births including infections, newborn respiratory distress syndrome, and the associated need for hospitalization in a neonatal intensive care unit, intraventricular hemorrhage, necrotizing enterocolitis, bronchopulmonary dysplasia, retinopathy of prematurity, or even death [11,13,14,15,16,17,18,19].

The aim of the study was to determine the effects of placental angiogenesis markers on adverse neonatal outcomes in mothers with preeclampsia. Additionally, the impacts of other parameters on adverse neonatal outcomes were assessed.

## 2. Patients and Methods

Our retrospective study included 77 newborns and their mothers diagnosed with preeclampsia hospitalized between 2018 and 2020 in the Pomeranian Medical University’s Department of Obstetrics and Gynecology and Department of Neonatology and Neonatal Intensive Care of the Autonomous Public Teaching Hospital no. 2 in Szczecin, Poland.

The criteria for preeclampsia diagnosis, as defined by the ISSHP (International Society for the Study of Hypertension in Pregnancy), were the occurrence after 20 wkGA of systolic blood pressure ≥140 mm Hg or diastolic blood pressure ≥90 mm Hg and proteinuria defined as daily protein loss >300 mg, or where no proteinuria was found then the occurrence of ≥1 of the following criteria:(1)Hematological disorders (thrombocytopenia, DIC, hemolysis),(2)Serum creatinine content >1.1 mg/dL or a 2-fold increase in its baseline level where no other kidney disease found observed,(3)Increased serum liver enzymes ≥2 times the upper limit of the standard or severe right upper quadrant or epigastric pain,(4)Neurological signs or visual impairment,(5)Pulmonary edema,(6)Intrauterine growth restriction.

5-mL venous blood samples were collected in tubes without an anticoagulant from preeclamptic mothers and subsequently sFlt-1 and PlGF concentrations were determined using the Cobas e801 (Roche Diagnostics) analyzer.

For each newborn, their birth week, sex, delivery method, 5-min Apgar score, and basic anthropometric measurements such as neonatal birth weight, were assessed. The Fenton growth charts (www.ucalgary.ca/fenton, accessed on 1 May 2022) were used to determine the birth weight percentiles. Each newborn was evaluated for length of hospitalization and for adverse neonatal outcomes such as:(1)Congenital or late-onset infections—diagnosed based on clinical signs, laboratory test results, and/or blood cultures,(2)Respiratory distress syndrome (RDS)—diagnosed based on clinical signs and a chest X-ray,(3)Patent ductus arteriosus (PDA)—diagnosed based on echocardiography and clinical signs,(4)Necrotizing enterocolitis (NEC)—diagnosed based on clinical signs and pathological radiological signs,(5)Intraventricular hemorrhage (IVH)—diagnosed based on brain ultrasound results(6)Retinopathy of prematurity (ROP)—diagnosed based on ophthalmological examination results,(7)Bronchopulmonary dysplasia (BPD)—diagnosed based on oxygen dependence persisting beyond 28 days of age,(8)Death of the infant.

## 3. Statistical Analysis

The results of the study were statistically analyzed. Non-parametric Mann–Whitney U tests for quantitative data, and the chi-squared test or Fisher’s exact test for qualitative data, were used for the calculations. In addition, an analysis was performed comprising multiple logistic regression and an area under curve (AUC) calculation. Statistica ver. 13 software (StatSoft, Kraków, Poland) was used for the analysis.

## 4. Results

77 newborns and their mothers diagnosed with preeclampsia were included in the analysis. Our cohort main characteristics are shown in Table 1. 22 (28.6%) of the infants had at least one of the neonatal adverse outcomes listed above. 2 neonatal deaths were recorded. Almost all of the pregnancies ended in a C-section operation (95%). The most common indication for C-section was where there was a threatened eclampsia or the KTG showed abnormalities at labor induction.

For the sFlt-1/PlGF ratio, the cutoff point was established using the ROC curve. The cutoff point for the occurrence of any adverse neonatal outcome in our study population was 204 with a 64% sensitivity and 75% specificity, while the AOC was 0.644.

Table 2 shows a comparison of the infants’ perinatal outcomes and the neonatal complications for the sFlt-1/PlGF ratio cutoff point defined using the ROC curve. High sFlt-1/PlGF ratio neonates were born statistically significantly earlier and their birth weight was more often <10th and <3rd percentiles for the given gestational age. Additionally, their Apgar scores were lower. The newborns born to mothers with high ratios had longer hospitalization times and, generally, were more frequently diagnosed with any of the aforementioned adverse neonatal outcomes. However, the only statistically significant result was in the case of bronchopulmonary dysplasia. For the other complications listed above, the incidence was higher but was not statistically significant, which was probably the result of an insufficient size of the cohort.

Looking for the variables that had the most significant effects on the incidence of adverse neonatal outcomes, we used regression analysis to evaluate such parameters as the sFlt-1/PlGF ratio, birth week, birth weight, maternal chronic arterial hypertension, and preeclampsia (either eo-PE or lo-PE). Univariate logistic regression shows statistical significance for particular parameters everywhere, although multivariate logistic regression showed that the single most important parameter predicting adverse neonatal outcomes was the birth week: 0.48(0.35–0.66) (OR (95% CI). For others, no statistical significance was achieved using multivariate logistic regression. Therefore, using the ROC curve, the cutoff point for the latter parameter was determined. The cutoff point for the occurrence of an adverse neonatal outcome was 32 weeks with a sensitivity of 77% and specificity of 95%, while the AOC was 0.91. Figure 1 shows a comparison between the ROC curves for the sFlt-1/PlGF ratio and the birth week.

Table 3 contains a comparison between the infants’ perinatal outcomes and the neonatal complications for the birth week cut-off point defined using the ROC curve. Neonates with lower birth week values had higher sFlt-1/PlGF ratios. They also had worse Apgar scores. Moreover, their hospitalization times were longer and, in general, they were more frequently diagnosed with any of the above-listed adverse neonatal outcomes. In fact, for all of the listed complications these frequencies were statistically significantly higher. Only for necrotizing enterocolitis (NEC) was no statistical significance reached, which was probably the result of an insufficient size of the cohort.

Table 4 contains a comparison between the perinatal outcomes and neonatal complications in infants born prior to or at 32 wkGA. Additionally, they are divided between those with a higher and a lower sFlt-1/PlGF ratio. The neonates born to mothers with higher sFlt-1/PlGF ratios were statistically significantly more common to be diagnosed with any of the aforementioned adverse neonatal outcomes. Also, the incidence of the particular complications was higher in them, although not reaching statistical significance (probably due to the cohort being too small).

## 5. Discussion

The present study assessed infants born to mothers with either early- and late-onset preeclampsia. We proved that the sFlt-1/PlGF ratio has an impact on the incidence of adverse neonatal outcomes. However, we also showed that the birth week remains the most important indicator of the occurrence of specific neonatal complications. One of the greatest challenges faced by contemporary perinatal medicine is to limit the incidence of preterm birth. As multiple authors have shown, maternal PE lowers the age at which the child is born. Unfortunately, there is currently no treatment available that would significantly extend the duration of gestation after PE diagnosis [20,21,22]. This study shows that an increased sFlt-1/PlGF ratio in the mother correlates with an increased incidence of preterm labor and body weight <10th and <3rd percentiles for the gestational age. This is most likely due to reduced uteroplacental flows and consequently placental ischemia, which may result in intrauterine growth restriction and necessitate of ending of pregnancy earlier [14,23,24,25].

Notably, preterm birth is one of the most important causes of complications during the neonatal period, such as infection, respiratory distress syndrome (RDS), retinopathy of prematurity, intraventricular hemorrhage (IVH), necrotizing enterocolitis (NEC), and death. The development of complications in the form of respiratory distress syndrome and bronchopulmonary dysplasia (BPD) may be linked to the pathophysiology of preeclampsia, where the number of antiangiogenic factors (sFlt-1) is increased and the concentrations of angiogenic factors (PlGF and VEGF) are reduced. Previous research shows that abnormal angiogenesis present in PE may cause impaired development of pulmonary vessels in the fetus and therefore contribute to the development of respiratory distress syndrome [10,11,12,26]. Authors have tested tracheal aspirate collected from preterm infants and found lower VEGF concentrations in samples from children born to preeclamptic mothers [27]. Other authors who studied a very large population of children born between 28 and 42 wkGA showed that maternal arterial hypertension and preeclampsia increased the risk of respiratory disorders in preterm infants and infants born at term alike [28]. There have been reports claiming that maternal PE increases the risk of NEC preterm infants [13]. Researchers have also attempted to link the occurrence of neonatal infections to preeclampsia, but have been unable to find evidence of their higher prevalence compared to infants born to healthy mothers [29]. The present study shows a higher incidence of general adverse neonatal outcomes and bronchopulmonary dysplasia in infants born to mothers with high sFlt-1/PlGF ratios for the whole study population, which confirms the aforementioned reports. As for the particular complications, although more frequent indeed they did not reach statistical significance (Table 2). Additionally, as shown in Table 4, adverse neonatal outcomes were also significantly more common in neonates born prior to a full 33 wkGA if the sFlt-1/PlGF ratio was higher. In this case, too, the particular complications were more common, although they did not reach statistical significance. Perhaps a larger cohort would have allowed for such a correlation to be determined. Importantly, preeclampsia, especially its early-onset form, is a rare complication which calls for multicenter research into this disorder.

There are reports that preeclampsia is linked to longer hospitalizations of neonates in the neonatal intensive care units (NICU) [19]. Other authors have shown that a longer stay in the neonatal care unit may be related to an elevated sFlt-1/PlGF ratio or to an increased sFlt concentration or a decreased PlGF content alone [30,31]. In our study, the sFlt-1/PlGF ratio demonstrated similar correlations, with the neonates born to mothers with higher ratios spending more time in hospital.

In our study population, the observation of infants born to preeclamptic mothers focused on their time spent in hospital, although the long-term complications suffered by these children must not be disregarded. The literature suggests that children of mothers with preeclampsia have an increased risk of depression in adult life, although maternal preeclampsia results in a reduced risk of developing severe mental disorders in male offspring [32,33]. In addition, infants born at term to PE mothers carry an increased risk of developing endocrine and metabolic diseases [34].

In that case, the question arises concerning what should be done to prevent infants born to preeclamptic mothers from developing complications. The answer is that acetylsalicylic acid therapy should be introduced. The authors of the ASPRE study showed that administering acetylsalicylic acid to women carrying the risk of early-onset preeclampsia (i.e., the one developing before a full 34 wkGA), reduced that risk by up to 95% [35,36]. The same authors noticed significantly shorter hospitalizations in neonatal intensive care units for infants born to preeclamptic mothers who were using acetylsalicylic acid [37].

## 6. Conclusions

Infants born to preeclamptic mothers continue to pose a serious challenge in developed countries. This is especially true for children born to mothers with the early-onset form of the condition. The sFlt-1/PlGF ratio can be a useful tool in clinical practice in predicting short-term adverse neonatal outcomes, although we showed that the birth week is the most important factor determining the occurrence of such complications. Infants born prior to a full 33 wkGA, who were demonstrating adverse neonatal outcomes, had higher sFlt-1/PlGF ratios. Practically speaking, infants born beyond a full 33 wkGA developed no serious neonatal complications. This is why from clinical point of view, as mentioned earlier, prevention can be crucial, as it can help us significantly reduce the number of severely sick newborns. This will have a great impact on whether or not early-onset, as well as late-onset, complications will develop in these children in their childhood or adult lives. Perhaps a multicenter study on a larger cohort would help develop a clinically useful model for predicting adverse neonatal outcomes using the sFlt-1/PlGF ratio.

## Figures and Tables

**Figure 1 biomedicines-10-02048-f001:**
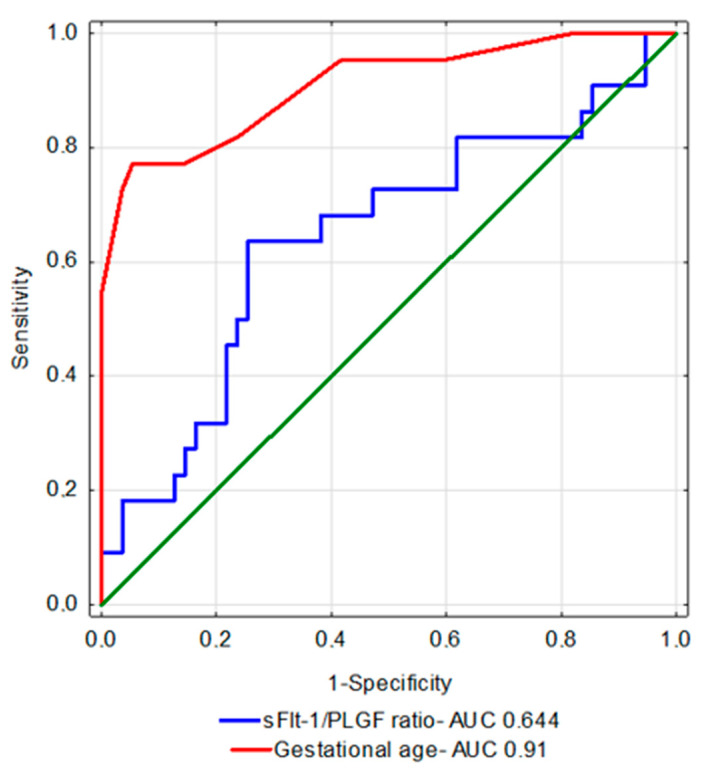
ROC curves for the selected parameters.

**Table 1 biomedicines-10-02048-t001:** Main characteristics of the cohort.

sFlt-1/PlGF ratio	153.7 (3.5–1460.5)	duration of hospitalization (days)	6 (2–87)
Gestational age (weeks)	35 (26–39)	adverse neonatal outcome	22 (28.6%)
birth weight (g)	2250 (510–4690)	congenital infection	15 (19.5%)
birth weight < 10 pc	20 (26%)	late-onset infection	7 (9.1%)
birth weight < 3 pc	9 (11.7%)	ROP	3 (3.9%)
5-min Apgar score	9 (6–10)	IVH	6 (7.8%)
Sex (male)	32 (41.5%)	NEC	1 (1.3%)
chronic hypertension	13 (16.9%)	RDS	14 (18.2%)
eo-PE	32 (41.5%)	PDA	5 (6.5%)
lo-PE	45 (58.5%)	BPD	8 (10.4%)
C-section	73 (95%)	death of the infant	2 (2.6%)

The quantitative variables are shown as the median (min-max). For the qualitative variables, are counts and percentages.

**Table 2 biomedicines-10-02048-t002:** Perinatal outcomes and neonatal complications depending on the sFlt-1/PlGF ratio.

	sFlt-1/PlGF Ratio		
	≥204 (n = 27)	<204 (n = 50)	*p*
sFlt-1/PlGF ratio	338.7 (208.7–1460.5)	83 (3.5–203.8)	<0.001
Gestational age (weeks)	33 (26–38)	36 (27–39)	<0.001
birth weight (g)	1470 (510–2820)	2585 (855–4690)	<0.001
birth weight < 10 pc	13 (48%)	7 (14%)	<0.001
birth weight < 3 pc	7 (26%)	2 (4%)	0.01
5-min Apgar score	8 (7–10)	9 (6–10)	0.01
Sex (male)	11 (41%)	21 (42%)	ns
chronic hypertension	2 (7%)	11 (22%)	ns
eo-PE	18 (67%)	14 (28%)	0.001
lo-PE	9 (33%)	36 (72%)	0.001
C-section	26 (96%)	47 (94%)	ns
duration of hospitalization (days)	23 (2–87)	5 (2–76)	<0.001
any of the following neonatal adverse outcomes	13 (48%)	9 (18%)	0.005
congenital infection	8 (30%)	7 (14%)	ns
late-onset infection	5 (19%)	2 (4%)	ns
ROP	2 (7%)	1 (2%)	ns
IVH	4 (15%)	2 (4%)	ns
NEC	1 (4%)	0	ns
RDS	8 (30%)	6 (12%)	ns
PDA	3 (11%)	2 (4%)	ns
BPD	6 (22%)	2 (4%)	0.03

The quantitative variables are shown as the median (min-max). For the qualitative variables, are counts and percentages. ns—non significant

**Table 3 biomedicines-10-02048-t003:** Perinatal outcomes and neonatal complications depending on the birth week.

	Gestational Age		
	≤32 (n = 20)	>32 (n = 57)	*p*
sFlt-1/PlGF ratio	232.7 (8.1–1460.5)	105.9 (3.5–949.7)	0.006
Gestational age (weeks)	30 (26–32)	36 (33–39)	<0.001
birth weight (g)	1070 (510–1915)	2570 (1230–4690)	<0.001
birth weight < 10 pc	8 (40%)	12 (21%)	ns
birth weight < 3 pc	4 (20%)	5 (9%)	ns
5-min Apgar score	8 (6–10)	10 (7–10)	<0.001
sex (male)	8 (40%)	24 (42%)	ns
chronic hypertension	5 (25%)	8 (14%)	ns
eo-PE	20 (100%)	12 (21%)	<0.001
lo-PE	0	45 (79%)	<0.001
C-section	19 (95%)	54 (95%)	ns
duration of hospitalization (days)	45 (2–87)	5 (2–23)	<0.001
any of the following neonatal adverse outcome	17 (85%)	5 (9%)	<0.001
congenital infection	12 (60%)	3 (5%)	<0.001
late-onset infection	6 (30%)	1 (2%)	<0.001
ROP	3 (15%)	0	0.02.
IVH	6 (30%)	0	<0.001
NEC	1 (5%)	0	ns
RDS	12 (60%)	2 (4%)	<0.001
PDA	5 (25%)	0	<0.001
BPD	8 (40%)	0	<0.001

The quantitative variables are shown as the median (min-max). For the qualitative variables, are counts and percentages. ns—non significant

**Table 4 biomedicines-10-02048-t004:** Perinatal outcomes and neonatal complications in infants born prior to 32 wkGA divided between those below and above the sFlt-1/PlGF ratio of 204.

	Gestational Age		
	≥204 (n = 13)	<204 (n = 7)	*p*
sFlt-1/PlGF ratio	313.3 (208–1460.5)	94.7 (8.1–203.8)	<0.001
Gestational age (weeks)	28 (26–31)	31 (27–32)	0.12
birth weight (g)	1035 (510–1850)	1200 (855–1915)	0.17
birth weight < 10 pc	6 (46.1%)	2 (28.6%)	ns
birth weight < 3 pc	4 (30.1%)	0	ns
5-min Apgar score	8 (7–9)	8 (6–10)	ns
sex (male)	3 (23.1%)	5 (71.4%)	ns
chronic hypertension	2 (15.4%)	3 (42.9%)	ns
C-section	13 (100%)	6 (95%)	ns
duration of hospitalization (days)	51 (27–87)	27 (2–76)	0.15
any of the following neonatal adverse outcome	13 (100%)	4 (57.1%)	0.01
congenital infection	8 (61.5%)	4 (57.1%)	ns
late-onset infection	5 (38%)	1 (14.3%)	ns
ROP	2 (15.3%)	1 (14.3)	ns
IVH	4 (30.7%)	2 (28.6%)	ns
NEC	1 (7.7%)	0	ns
RDS	8 (61.5%)	4 (57.1%)	ns
PDA	3 (23%)	2 (28.6%)	ns
BPD	6 (46%)	2 (28.6%)	ns

The quantitative variables are shown as the median (min-max). For the qualitative variables, are counts and percentages. ns—non significant

## Data Availability

The data presented in this study is available upon request from the author for correspondence. The data is not publicly available, as not all patients agreed to publicly disclose the data.

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
