# Peer review of "Adverse Neonatal Outcome of Pregnancies Complicated by Preeclampsia"

_biomedicines, 2022, doi:10.3390/biomedicines10082048_

Round 1
Reviewer 1 Report
This is an interesting study focused on the evaluation of infants born to preeclamptic mothers. The research is well-conducted and the results of great relevance from a clinical point of view.
Nevertheless, some minor questions must be addressed before its acceptance for publication:
1. Please, refer to "cohort" instead of "group".
2. Please, revise Table 3 foot.
3. Please, add a conclusion paragraph clearly indicating the relevance of the results obtained from a clininacl point of view.
Author Response
All the authors sincerely appreciate all the comments that enhance our work and, above all, help us improve our professional activity.
We've refered to cohort instead of group and also revised Table 3 foot.
We've addeed a conclusion paragraph and indicated the relevance of the results from clinical point of view.
We appreciate again all your suggestions that do nothing but help us improve.
Thank you very much
Reviewer 2 Report
In this manuscript authors assessed the perinatal outcomes of infants from PE pregnancies. They found that infants born from mothers with high sFlt-1/PlGF ratios had longer hospitalization times and, generally, were more frequently subject to adverse neonatal outcomes.
The manuscript is interesting but some points must be improved. In particular:
Introduction: It deserves to be highlighted that PE pregnancies are also characterised by trophoblast immaturity (PMID: 32529396). This is a very important feature of this disease since it may be the cause of the alteration of many factors analysed by the authors (in particular the PlGF).
Table 1: The first two columns must be separate from the other two in order to make it easier to understand. Moreover, unit of measurement must be reported. E.g Gestational age (weeks), birth weight (g)....
Figure 1: Image quality must be improved and AUC must be shown in the figure as well.
Author Response
All the authors sincerely appreciate all the comments that enhance our work and, above all, help us improve our professional activity.
We've highlighted in the introduction that PE pregnancies are also characterised by trophoblast immaturity.
We've corected the tables in the manuscript and made FIgure 1 more clearly.
We appreciate again all your suggestions that do nothing but help us improve.
Thank you very much